# Effects of Fe-DTPA on Health and Welfare of the African Catfish *Clarias gariepinus* (Burchell, 1822)

Marc-Christopher Hildebrand [1,*], Alexander Rebl [2], Julien Alban Nguinkal [2], Harry Wilhelm Palm [1] and Björn Baßmann [1]

1    Department of Aquaculture and Sea-Ranching, Faculty of Agricultural and Environmental Sciences, University of Rostock, Justus-von-Liebig-Weg 6, 18059 Rostock, Germany
2    Fish Genetics Unit, Institute of Genome Biology, Research Institute for Farm Animal Biology (FBN), 18196 Dummerstorf, Germany
*    Correspondence: marc-christopher.hildebrand@uni-rostock.de; Tel.: +49-(0)381-498-3745

**Abstract:** Fingerlings (0.23 g) and juveniles (267.04 g) of African catfish (*Clarias gariepinus*) were reared for 32 days under experimental aquarium conditions and were exposed to either 0.75 mg/L or 3.0 mg/L diethylenetriaminepentaacetic acid-iron(II) (Fe-DTPA) and 3.0 mg/L or 12.0 mg/L Fe-DTPA in the water, respectively. These treatment groups were compared to a control group without additional Fe-DTPA. The growth, mortality, ethological indicators (activity, agonistic interactions, air-breathing), leukocyte distribution, histopathological changes in liver and gills, and genetic biomarkers were evaluated for each group. While the growth, mortality, and behavior were not significantly different between the groups, the lymphocyte count in the fish's blood increased significantly in all groups during the course of the experiment, but independently from the treatments. A similar trend ($p > 0.05$) was observed in monocytes. The number of granulocytes decreased significantly, but independently from the treatments. These changes indicated the possibility of an ongoing immune response in the fish from all treatments that might be caused by the increasing aggressive behavior of the fish. However, the Fe-DTPA treatments did not cause a notable suppression or enhancement of the immune reactions. $Fe^{3+}$ accumulations in liver tissues were detected at the tested concentrations, and further changes occurred in the cells of the gills. Gene-expression biochips were used to simultaneously quantify the transcript levels of 34 genes associated with iron metabolism and stress physiology in head kidney samples. The obtained gene-expression profiles did not reveal any significant differences across either the different treatments or the time points. The results indicate that Fe-DTPA supplementation in the tested concentrations can be considered relatively harmless for the health and welfare of African catfish.

**Keywords:** aquaponics; fertilizer; fish histopathology; fish immunology; toxicity

## 1. Introduction

Nowadays, fish come into contact with various fertilizer components such as organic acids or metal compounds (often bound in chelate complexes) from different sources. These can be wastewater from the chemical industry, fertilizer inputs from agriculture or emissions from mining activities into natural water bodies or outdoor aquacultures.

In the combined farming of fish and plants, known as aquaponics, the addition of fertilizers can provide the plants with all the essential nutrients for optimum plant growth and thus increase yield. Aquaponics is generally considered a very sustainable production method [1] as it reuses the nutrient-rich water from aquaculture to grow plants. In particular, nitrate and phosphate can be used in a resource-efficient way, and water can be saved [2–4]. However, the linkage to aquaculture cannot provide all the essential nutrients for the plants, so supplementing with selective fertilizers is often required [1,4]. Plants grown in aquaponics have species-specific requirements for macro- and micro-nutrients. The main

nutrients (nitrogen and phosphorus) are mainly supplied to the crop plants through the aquaculture unit. However, the supply of important micronutrients (such as zinc, iron, manganese, magnesium, calcium, potassium, and sulfur) often remains at a level that is insufficient for the plants [5]. In decoupled aquaponic systems, where the recirculated water is not directly returned to the fish in the aquaculture unit, the required nutrients are often added by spraying them onto the plants [6,7], and the residues in the water are later discarded as the uncertain toxic properties appear to make it unusable for fish farming. This, however, contradicts the sustainability idea of an otherwise circulating aquaponic system with low water consumption. In recent years, however, motivations to better adjust fertilizer concentrations in aquaponic systems to suit both of the production components (fish and plant), and thus further optimize production toward economic efficiency and sustainability, have repeatedly emerged [1,8,9].

Nowadays, a wide range of different trace nutrients is used as fertilizers in aquaponic systems such as $Fe^{3+}$, $Cu^{2+}$, $B^{3+}$, $Mo^{3+}$, $Mn^{2+}$, $Zn^{2+}$, and $K^+$. These can be added to the water either as salts ($FeSO_4$, $KCl$, and $KSO_4$), chelated compounds (Fe-EDTA, Fe-EDDHA, and Fe-DTPA), or as component in organic acids (Fulvic acids, Fe-arginine, Fe-glycine, and Fe-histidine) [4,8,10–13]. These supplements are often derived from hydroponic fertilizers.

However, unbalanced ratios of nutrients that are important for the plants can also have a negative effect on the health and growth of the fish. Therefore, targeted fertilization and a minimum of control of the nutrient concentrations in the system are crucial. In particular, the concentration limits of micronutrients should be known for the stocked fish species and should not be exceeded. Overdoses of micronutrients can lead to damages of the internal organs, blood diseases, reduction in oxygen uptake capacity, changes in behavior, and even increased mortality. Recent studies on this topic report different negative responses of fish, such as increased mortalities (lethal dose 50% mortality, LD50 values) to Fe, Cu, Zn, Ni supplements, such as protein digestion, behavioral responses, alteration in white blood cell counts, reductions in hemoglobin, and mean corpuscular hemoglobin or pathological alterations in the organs [8,10–12,14].

For the African catfish (*Clarias gariepinus* Burchell, 1822), tolerance tests have already been carried out on compatibility with nitrate-phosphorus-potassium (NPK) fertilizers, potassium trace nutrients, and iron sulphates [13,15,16]. Iron (Fe) plays a fundamental role in cell metabolism and is an essential nutrient for both fish and plants. Fe exists in two different oxidative stages ($Fe^{2+}$ and $Fe^{3+}$). $Fe^{2+}$ is rather unstable and is more frequently present in acidic, reducing conditions, while $Fe^{3+}$ is predominantly present at alkaline pH [17,18]. $Fe^{3+}$ accumulates mainly at the bottom of a rearing tank or in depressions and is not found at the water surface as it is mainly a precipitation product. In finfish, Fe homeostasis is strictly controlled by uptake, as no regulated excretion mechanism for Fe in the body is known. Its insolubility makes $Fe^{3+}$ inaccessible for uptake by fish through the gills, so it is mainly obtained via diet [19–23]. The uptake of $Fe^{2+}$ from the water by the gills is negligible in most fish, as recent evidence has suggested [20,23]. $Fe^{2+}$ transport via a chelate complex (DTPA, diethylenetriaminepentaacetic acid) has also not been excluded [23–25]. Fe uptake in fish relies on many physiological mechanisms and is dependent on extrinsic as well as intrinsic factors that regulate the level required in the body and avoid excess as well as harmful effects [23,26].

In hydroponics, Fe is added in concentrations up to 5.0 mg/L to promote plant growth [5]. A targeted addition of Fe in combination with DTPA, which grants stability, protects the $Fe^{2+}$ of oxidation to insoluble $Fe^{3+}$ and thus enables availability of Fe in the water dependent on the pH value [27]. This also benefits plants as $Fe^{2+}$ ions can better dissolve on the rhizoplane of plant roots, thereby improving plant nutrient uptake [9,28,29].

In aquaponics, the Fe provided by the fish feed and feces is not sufficient to meet the nutrient requirements for optimal plant growth and health (2–5 mg/L) [9,30–32]. According to Strauch et al. [33], the Fe concentration in the process water from African catfish is very low.

The current study aims to evaluate the effects on health and welfare of the African catfish to non-lethal but aquaponic-related concentrations of the chemical chelator agent Fe-DTPA. We assume that Fe-DTPA in concentrations that meet the minimum requirements for aquaponic systems (approx. 4 mg/L) do not have a negative effect on the health and welfare of African catfish. With this assessment, lower concentrations of Fe-DTPA could be therefore used for fertilizer supplementation in coupled aquaponic systems. African catfish fingerlings and juveniles were challenged with different concentrations of Fe-DTPA with an emphasis on modern health and welfare responses. Growth performance, ethological indicators, blood smears, histopathological analysis, and expression of critical genes were recorded to assess possible hazards of Fe fertilization to the fish.

## 2. Materials and Methods

### 2.1. Experimental Design and System Maintenance

Two experiments were conducted at the aquaculture research facility 'Fish glass house' of the University of Rostock, in northern Germany. Each experiment lasted for 32 days. The first experiment (referred to as Exp. A) was designed for fingerlings, whereas the second experiment (referred to as Exp. B) was designed for juveniles. Both the experiments consisted of three separate recirculation systems, each with three tanks (Exp. A: L × W × H: 100 cm × 50 cm × 32 cm; Exp. B: L × W × H: 100 cm × 50 cm × 35 cm) and a filter unit (approx. 450 L each). This resulted in total volumes of 950 L in Exp. A and 1000 L in Exp. B. In Exp. A, the volume of the tanks was limited to about 50 L, each with vertically placed filter mats because of the small number of fish (see below). Exp. B had no restrictions on tank volumes.

To achieve the targeted concentrations of Fe, we used Fe-DTPA containing 7% mass of $Fe^{2+}$ provided by Phygenera Germany (EG No.243-136-8). The fingerlings of Exp. A were exposed to treatments of either 0.75 mg/L (referred to A-FeDTPA-0.75) or 3.0 mg/L Fe-DTPA (referred to A-FeDTPA-3), whereas the juveniles of Exp. B to treatments of either 3.0 mg/L (referred to B-FeDTPA-3) or 12.0 mg/L Fe-DTPA (referred to B-FeDTPA-12). The fertilizer concentration was lower for fingerlings (Exp. A) because they were more vulnerable. The treatment groups in each of the experiments were compared with a control group without Fe-DTPA addition (referred to FeDTPA-0) and with a total iron concentration of 0.062 mg/L.

In order to keep the concentrations of Fe-DTPA in the water stable, a dosing computer (Profilux4, GHL) and a dosing pump (Doser3, GHL) were used. For adjustment, a prepared Fe-DTPA solution was slowly added to the water cycle within the sump. Water exchange was performed after 2 weeks, replacing 10% of the total volume.

The respective needs of its mass were calculated by a modified formular according to Latimer [34]:

$$\frac{C_w \cdot D_f}{F \cdot C} \cdot V_k = m \tag{1}$$

$C_w$ = targeted concentration (mg/L)
$D_f$ = dilution factor (1)
$F$ = percentage element concentration of the fertilizer (%)
$C$ = conversions constant from imperial to the metric system (mg)
$V_k$ = total processed volume (L)
m = fertilizer mass (g).

The volumetric standard solutions (3000 and 6000 mg/L) were assessed to initiate the fertilizer solution of the treatments. The respected volumes were then calculated by using a known dilution formula (see below). The resulting volumes were processed by the dosing unit with an error of 1.0 mL per 500 mL.

$$C_1 \cdot V_1 = C_2 \cdot V_2 \tag{2}$$

$C_1$ = concentration of standard solution (mg/L)

$V_1$ = volume of the standard solution (L)
$C_2$ = targeted concentration in the system (mg/L)
$V_2$ = needed intake of standard solution (L).

The water quality parameters (temperature, pH, dissolved oxygen (DO), electric conductivity (EC), and redox potential) were measured daily using a portable multimeter (HQ40, Hach Lange). In a three-day interval, water samples were taken from the sumps to analyze nitrate ($NO_3^-$), nitrite ($NO_2^-$), ammonium ($NH^{4+}$), and ortho-phosphate ($PO_4^{2-}$) using an autoanalyzer (Gallery™, Automated Photometric Analyzer, Thermo Fisher Scientific, Waltham, MA, USA). The total Fe concentrations were determined using Hach-Lange cuvette tests (LCK321 & LCK521) and by performing acid digestion (LCW902) via a separate photometer (DR3900 RFID Spectrophotometer, Hach-Lange, Iowa Ames, IA, USA). If the iron concentrations differed from the targeted concentrations, additional Fe-DTPA solution was added.

### 2.2. Fish Stocking and Feeding

For both the experiments, 288 African catfish were obtained from a local fish hatching company (PAL Aquaculture GmbH, Kiel, Germany). The fingerlings of Exp. A were stocked with 14 fish per 50 L (3.6 fish/L), with mean weights of 0.23 ± 0.04 g and mean lengths of 3.0 ± 0.14 cm, into the nine tanks. The number of experimental animals was reduced as much as possible so that few animals would be stressed. The juveniles of Exp. B were stocked with 18 fish per 120 L (6.7 fish/L), with mean weights of 267.04 ± 54.93 g and mean lengths of 31.0 ± 2.94 cm, into the nine tanks. All the fish were acclimated for 9 days in untreated water before they were exposed to the different concentrations of Fe-DTPA.

During the experiment, the fish were fed using a commercial catfish diet (Exp. A: Coppens Start Premium 0.3–1.5 mm; Exp. B: Coppens Special Premium 5.0 mm). To avoid high nutrient loads, the calculated feed amounts were set for fingerlings with 3% and for the juveniles with 0.7% of their mean bodyweight, respectively. The corresponding daily feed amount was divided by the number of feeding events per day since the fingerlings were fed four times (at 0, 6, 12, and 18 pm), whereas the juveniles were fed only once (at 12 pm). In Exp. A, during the experimental period of 32 days, 13 g per tank were fed; in Exp. B 60 g were fed.

The feed conversion ratio (FCR) was calculated with:

$$\frac{TFI}{(W_1 - W_0)} = FCR \tag{3}$$

$TFI$ = total feed intake (g)
$W_0$ = initial fish weight (g)
$W_1$ = final fish weight (g).

### 2.3. Sampling

In Exp. A, all the fish in each tank were caught and subsequently weighed and had their lengths measured on days 11, 12, 21, and 32 to monitor their weekly growth performance. Afterwards, the fish were returned to their respective rearing tanks. In Exp. B, on days 11, 12, 21, and 32, three juvenile fish were taken from each group, stunned by brain percussion and killed by puncture of the heart. The weight and length of each individual fish was measured. Fish sampled on days 11, 12, 21 and 32 were additionally investigated for their possible genetical response to Fe-DTPA, and on days 11, 21, and 32, for alterations in their blood. After 32 days, as each experiment ended, all individuals were counted, measured, and killed.

Fulton's condition factor (CF) was estimated regarding to Ricker [35]:

$$\frac{100 \cdot W}{L^3} = CF \tag{4}$$

CF = Fulton's condition factor

$W$ = weight (g)

$L$ = length (cm).

All treatments were carried out in accordance with the EU guidelines for animal experiments and were approved by the relevant ethics committee.

### 2.4. Blood Smear Analysis and Skin Lesion Documentation

In Exp. B, on days 11, 21, and 32, blood samples of approx. 5 mL were taken from the caudal vessels of the fish and transferred to a cooled EDTA tube (BD Vacutainer, K2E 5.4 mg). Three hours later, blood smear preparations were created from 50 μL of blood and dyed according to Pappenheim [36]. The samples were then fixed within Canada balsam in order to evaluate the immune fitness by cell type distribution by a leucogram. These preparations were analyzed under a light microscope (Olympus BX 53, application software Cellsens, v. 1.5) using a 500× magnification. To arrange a differential blood count for a specific group, 100 cells per blood smear were counted, and the leucocyte cell types were determined. This process was repeated five times; according to Rümke et al. [37], this increased the statistical accuracy. The means of the respective groups were then calculated, which included the cell counts of three samples from a given group.

Neutrophile cell classification was conducted according to Ainsworth et al. [38], who had reported these classifications in various fish species, including African catfish. Precursor cells were determined according to Clauss et al. [39]. From the three types of granular leucocytes found in mammals, only neutrophils and eosinophils are found in *C. gariepinus*. The series of neutrophilic granular leucocytes consist of the neutrophilic progranulocytes, neutrophilic meso-granulocytes, neutrophilic meta granulocytes, and mature and aged neutrophilic granulocytes. The proportion development stages for neutrophilic granulocytes allows a small look into an upcoming inflammation reaction of a fish [39]. Other leucocyte types (lymphocytes, monocytes, and eosinophile) were determined according to Boomker [40].

In Exp. B, the number and area of skin lesions were recorded for each sampled fish. To determine the lesion area, a clear template with a 0.25 cm$^2$ grid was placed over each lesion and the smallest possible lesion area was recorded. All lesion areas per fish were then summed. In Exp. A, this was not performed because the fish were too small in size.

### 2.5. Histological Analysis

When sampling the fish in Exp. B, special attention was paid to different organs to investigate toxicity. Liver tissue was dissected by cutting off a 5 mm$^3$ cube from the middle area of the liver. Gill tissue (1 cm) was taken from the outer left branchial arch. Both the organs were preserved in 15% formalin/5% methanol and stored at 8 °C. In addition, the upper part of the head kidney from days 11, 12, 21, and 32 was dissected and stored at −80 °C for gene expression analysis. The tissue samples of liver and gill samples were washed and fixed in a histokinette by 100% ethanol and an embedding agent (Histosec 56–58, Merk, New York, NY, USA), respectively. Preparations were then embedded in blocks of paraffin and cut into 1 μm thick cross-sections on a microtome. Subsequent staining solutions for Fe$^{3+}$ and Fe$^{2+}$ (Prussian blue and Turnbull's blue) were prepared according to Mulisch [36] and used in the dying process through an automated staining bench (ST5010-CV5030, Leica, Wetzlar, Germany). Crated histopathological samples were analyzed using a scanning microscope (Leica DM4 B, application software LAS X, v. 5.02). Images were assessed and severity grading was conducted according to Johnson et al. [41]. The following grades were defined: (G0), not remarkable (no findings associated with a particular diagnostic criterion); (G1), minimal (inconspicuous to barely noticeable, fewer than 2 occurrences per microscopic field); (G2), mild (noticeable feature of the tissue, 3–5 occurrences per microscopic field); (G3), moderate (dominant feature of the tissue, 6–8 occurrences per microscopic field); (G4), severe (overwhelming feature of the tissue, more than 9 occurrences per microscopic field).

*2.6. RNA Isolation, Primer Design and Multiplex Quantitative PCR*

A list of genes related to Fe homeostasis and general stress responses was identified based on a comprehensive literature research. In addition, we derived primer sequences from traditionally chosen reference genes (*rna18s*, *rpl*, *actb*, *gapdh*) [42] that proved stable transcript abundances across different treatment groups in previous investigations on fish [43]. The sequence information for the reference-gene orthologs in African catfish was available at the NCBI nucleotide database. To identify the orthologous target genes from African catfish we searched against our *C. gariepinus* genome (NCBI accession codes GCA_024256425, GCA_024256435, GCA_024256465) obtained with Oxford Nanopore PromethION, PacBio Sequel, Illumina NovaSeq and Illumina HiSeqIllumina RNA-seq technologies. Reciprocal BLASTs against the NCBI nucleotide database were performed confirming the identity of the obtained *C. gariepinus* sequence fragments. If we could not identify any ortholog sequence from *C. gariepinus*, the respective orthologs of close catfish relatives, including *Ictalurus punctatus*, *Pangasianodon hypophthalmus*, and *Silurus meridionalis*, were utilized. The verified sequence fragments were imported into Pyrosequencing Assay Design software (version 1.0.6; Biotage, Uppsala, Sweden) to predict optimal primer oligonucleotides specific for African catfish with a primer score $\geq$ 85 to amplify target fragments between 20 and 23 bp (Table 1). All primer-pair assays were tested on a diverse array of tissue samples (brain, gills, gonads, head kidney, intestine, liver, muscle, skin, and spleen) before measurement to validate the proper amplification of target-gene fragments. For these tests, the LightCycler 96 System (Roche, Mannheim, Germany) along with with the SensiFAST SYBR No-ROX Kit (Bioline/Meridian Bioscience, Memphis, TN, USA) was used. Melting curve analyses and agarose gel electrophoresis validated the amplification of specific PCR products with the targeted length. Thirty-eight assays passed the pre-tests (Table 1) and were used to profile the expression of selected genes in the samples from the above experiment.

The total RNA from the head kidneys of the treated and control fish (n = 3 per group) was extracted using 1 mL TRIzol Reagent (Thermo Fisher Scientific) and subsequently purified with the RNeasy Mini Kit (Qiagen, Hilden, Germany). Residual DNA was digested using the RNase-free DNase I (Qiagen) for 12 min at 37 °C. The concentration and the purity of the extracted RNA were assessed using the NanoDrop OneC spectrophotometer (NanoDrop Technologies, Wilmington, DE, USA).

The integrated fluidic circuit (IFC) technology from Standard BioTools Gene Expression biochips was used to profile the expression of 34 target genes and 4 reference genes (*rna18s*, *rpl*, *actb*, *gapdh*) in the extracted RNA specimens from all the Exp. B groups. The multiplex quantitative PCR (qPCR) analyses were performed on a 48.48 IFC chip (Standard BioTools, San Francisco, CA, USA) using the BioMark HD system (Standard BioTools). To this end, we adjusted the total RNA at a concentration of 10 ng/µL and reverse-transcribed 1 µL (42 °C, 30 min) using the Reverse Transcription Master Mix (Standard BioTools). The resulting cDNA aliquots were mixed with primers (100 µM) and the PreAmp master mix (Standard BioTools) and individually preamplified in 15 cycles (95 °C, 15 s; 60 °C, 4 min) in a TAdvanced thermocycler (Biometra, Göttingen, Germany). After this preamplification step, exonuclease I (New England BioLabs, Ipswich, MA, USA) was added to degrade single-stranded oligonucleotide primers. After a 30 min incubation period at 37 °C, 43 µL TE buffer (Sigma, Kawasaki, Japan) was added per sample. Each 50-µL-cDNA sample was diluted in SsoFast EvaGreen Supermix with Low ROX (Bio-Rad, Hercules, CA, USA) and 20×DNA Binding Dye Sample Loading Reagent (Standard BioTools) to produce the sample mixes. After priming the 48.48-IFC chips in the MX Controller (Standard BioTools), the primers and the sample mixes, along with one no-template (water) control, were transferred to the assay and sample inlets on two primed 48.48-IFC chips. Finally, multiplex qPCR was conducted following the manufacturer's thermal protocol 'GE Fast 48.48 PCR+Melt v2.pcl'.

**Table 1.** Oligonucleotide-primer sequences derived from *Clarias gariepinus*, *Ictalurus punctatus* or *Pangasianodon hypophthalmus*.

| Gene Symbol | Gene Product | Function | Sense Primer (5′→3′), Antisense Primer (5′→3′) | Source (Species; Accession Code) | Fragment Length (bp) | PSQ Score |
|---|---|---|---|---|---|---|
| **Reference genes:** | | | | | | |
| *rna18s* | 18S ribosomal RNA | Structure of eukaryotic ribosomes | CTCTGCTGGACGATGGCTTAC, TCGATGAAGAACGCAGCCAGC | *C. gariepinus*; GQ465239 | 94 | 100 |
| *actb* | Actin-beta | Cell structure and motility, intercellular signaling | ACCACCACAGCCGAGAGAGAA, CTTCCAGCCATCTTTCCTTGGT | *C. gariepinus*; EU527191 | 204 | 86 |
| *gapdh* | Glyceraldehyde-3-phosphate dehydrogenase | Carbohydrate metabolism | TATGAAGCCCGCTGAGATCCC, GCCTCTTCTCACTTGCAGGGT | *C. gariepinus*; AF323693 | 106 | 99 |
| *rpl* | Ribosomal protein, large subunit | Structure of eukaryotic ribosomes | ACTAAATAGCAACTGATCCCTATC, GAATATCTGACCACTAAGATCCG | *C. gariepinus*; MW080924 | 134 | 96 |
| **Target genes:** | | | | | | |
| *atp6v1g1* | ATPase H+ transporting v1 subunit g1 | Intercellular Fe homeostasis | CGGAAAAACCGCCGCTTGAAG, GACCAAGGAAGCCGCGGCAC | *P. hypophthalmus*; XM_026922532 | 106 | 84 |
| *c3a* | Complement component 3, variant a | Bacteria opsonization and destruction | ATGTCTTTCGATGTCACGGTTTAT, TCGAACCAAGAGTAACGGCATG | *I. punctatus*; XM_017457024 | 114 | 93 |
| *casp3* | Caspase 3 | Apoptosis | CTCTTTATCATTCAGGCTTGTCG, GTACTCTACTGCTCCAGGTTATT | *I. punctatus*; XM_017473312 | 139 | 95 |
| *casp8* | Caspase 8 | Apoptosis | GTTATCAGCCGAAGCCGCTCA, ATCCAGAGCTATGATGTGTCCG | *Cyprinus carpio*; XM_042730675 | 157 | 91 |
| *ccnb1* | Cyclin b1 | Control of the G2/M transition phase of the cell cycle | TCAAAAATCGGAGAGGTTACAGC, TGCACTTTGCTCCCTCTCTGG | *I. punctatus*; NC_030443 | 103 | 91 |
| *cp* | Ceruloplasmin | Copper transportation, oxidation of iron ($Fe^{2+}$ to $Fe^{3+}$) | CCACAACGTTCTAGAAGAATCATA, CTAAGAATGGAGGTCCAACTAAAA | *I. punctatus*; JF914943 | 155 | 87 |
| *cs* | Citrate synthase | Aerobic metabolism | GGTGGTGAAGTGTCCGATGAAA, GCTATGGGCATGCTGTCCTGA | *I. punctatus*; XM_017487510 | 94 | 94 |
| *hmox1a* | Heme oxygenase 1 | Cellular response to xenobiotic stimulus | GATTCTTCTGTGTTCCCTGTATG, CCATCTACTTCCCTCAGGAGC | *I. punctatus*; XM_017491622 | 104 | 95 |
| *hsf* | Heat-shock transcription factor 1 | ERK signaling, stress response | GTGCAGTCCATCAACTTTGATTC, CTATTCAGGAGTTGCTGTCAGAA | *I. punctatus*; XM_017455240 | 111 | 93 |
| *hsp90ab1* | Heat-shock protein 90 alpha family class b member 1 | Chaperone function, stress response | GAACATCAAGCTGGGCATCCAT, TTACTACATCACTGGTGAGAGCA | *I. punctatus*; XM_017456214 | 167 | 87 |
| *hspb1* | Heat-shock protein family b (small) member 1 | Differentiation of cell types, stress response | ACAGGACAACTGGAAGGTGAAC, GATTATCGGAAACCATGAGGAGA | *Clarias batrachus*; KT359728 | 107 | 97 |

**Table 1.** *Cont.*

| Gene Symbol | Gene Product | Function | Sense Primer (5′→3′), Antisense Primer (5′→3′) | Source (Species; Accession Code) | Fragment Length (bp) | PSQ Score |
|---|---|---|---|---|---|---|
| *hspd1* | Heat shock protein family d (hsp60) member 1 | Chaperone function, stress response | GCACGCTTGTCCTCAACAGGTT, AGACATGGCGATTGCTACTGGA | *I. punctatus*; XM_017469365 | 113 | 91 |
| *igf1* | Insulin-like growth factor 1 | Anabolism, growth | CGCCCAAAACACCAAAGAAACC, AGTGACGAGAAGAGGAGAGCG | *I. punctatus*; NM_001200295 | 164 | 100 |
| *il2* | Interleukin-2 | Activation and proliferation of lymphocytes | GTCGGCCTGGGAAAAAGCCAAT, TTATGTGTTTGCACCAGACAACG | *I. punctatus*; XM_017474923 | 162 | 95 |
| *il4* | Interleukin-4 | Activation and proliferation of leucocytes | ATGAATCCTTGTGGAAGATTAGAG, GGAGTATTTGGTGAGAGAGGTAA | *P. hypophthalmus*; XM_026924084 | 108 | 86 |
| *il6* | Interleukin-6 | Acute-phase response | GCAGTTGAAACGGGACTTCCCA, TGTACCAAGCTTACCTGCCCTA | *I. punctatus*; XM_017455306 | 162 | 96 |
| *kmt2a* | Lysine-specific methyltransferase 2a | Regulation of early development and hematopoiesis | ATTGGGTCGAAATCGTGCTGTAT, ATGATAAGTCTTCAGTGGCAGGT | *I. punctatus*; XM_017490460 | 121 | 90 |
| *mtf1* | Metal regulatory transcription factor 1 | Catabolic regulation of cartilages | GTAGGAGGGCATTCAGGGAAC, AGTCAGAACGCTGCCCCCTC | *I. punctatus*; XM_017475296 | 146 | 90 |
| *nkx2-3* | NK2 homeobox 3 | Cell differentiation | TACAGGACAACCTGGTGGAAAG, ACAACTCTTGGTTTCCTGCTCTT | *I. punctatus*; XM_017464595 | 119 | 89 |
| *nr3c1* | Nuclear receptor subfamily 3 group c, member 1, glucocorticoid receptor | Stress response | TGTAGAAGGCCAACACAACTATC, GAACCTAGAAGCACGCAAAAACA | *I. punctatus*; XM_017492397 | 137 | 94 |
| *oser1* | Oxidative stress-responsive serine-rich protein 1 | Oxidative stress | AACTGGCATGGATGCAGTCGAA, ACCTACTGTAGCTCTAAAATGCAA | *I. punctatus*; NM_001200453 | 120 | 93 |
| *osgin2* | Oxidative stress growth inhibitor | Oxidative stress | AGGAGCCTGGCATGCAATGGA, GTGACCAATGACCGGGCCAC | *I. punctatus*; XM_017467887 | 129 | 91 |
| *pgm3* | Phosphoglucomutase 3 | Carbohydrate metabolism | GACACAGGCAGGGCTGAATCT, CTTCGTACAGCACACTGTAACC | *I. punctatus*; XM_017494096 | 112 | 94 |
| *sirt1* | Sirtuin 1 | Oxidative stress | AGTGAGGTGCTAGGGTTAATGG, TTGGTTCTTATCGCTTTATTCAGC | *I. punctatus*; XM_017461869 | 148 | 91 |
| *slc30a5* | Solute carrier family 30, member 5 | Zinc transportation | AATAGTCACCAAAAGACAGTGGAT, CATCGTTGTGCTCGAACAACAG | *I. punctatus*; XM_017459891 | 134 | 90 |
| *slc39a8* | Solute carrier family 39, member 8 | Cellular zinc uptake, protection from inflammation-related injury and death | TTTAACCTGATCTCAGCCATGTC, TATGTTCCCTGAGATGAATGCCA | *I. punctatus*; XM_017489708 | 151 | 93 |
| *slc46a1* | Solute carrier family 46, member 1 | Folate transportation | AATGGCGACATGCACAAGGGTAT, AGAACAGCCTTGCCCCAGGG | *I. punctatus*; XM_017491375 | 129 | 88 |
| *sp1* | SP1 transcription factor | Cell growth, apoptosis, differentiation and immune responses | AGCACAGCAGGTGATCAGGGA, GAGAAGCGTGCACATGTCCATA | *I. punctatus*; XM_017450095 | 119 | 91 |
| *st8sia4* | St8 alpha-n-acetyl-neuraminide alpha-2,8-sialyltransferase | Synthesis of polysialic acid for cell adhesion molecule | GGTTCATGCAGTCAGAGGGTAC, CTTCTGCGATGAGATCCACTTG | *C. gariepinus*; PRJNA820763 | 112 | 85 |

**Table 1.** *Cont.*

| Gene Symbol | Gene Product | Function | Sense Primer (5′→3′), Antisense Primer (5′→3′) | Source (Species; Accession Code) | Fragment Length (bp) | PSQ Score |
|---|---|---|---|---|---|---|
| *stk39* | Serine/threonine kinase 39 | Stress response | TGTAGTTGTTGCTGCTAACCTTC, AGATCCCTGACGAGGTGAAGC | *I. punctatus*; XM_017469076 | 116 | 89 |
| *tlr5* | Toll-like receptor 5 | Detection of bacteria | GGCAGCATGGGAAAGGGAGTT, GTTAAGGCTCTGGATCTGTCCA | *I. punctatus*; NM_001200229 | 103 | 96 |
| *tnf* | Tumor necrosis factor alpha | Immune/acute-phase response | AAACCAGACGAGACCCAAGAAAT, TCTATGCAGTGGTTCGACAACG | *I. punctatus*; NM_001200172 | 130 | 96 |
| *ucp2* | Uncoupling protein 2 | Regulation of production of reactive oxygen species, function of mitochondria | GGCTCCAGATCCAAGGGGAGA, CCACGTAGTCTCTACAACGGG | *I. punctatus*; XM_017489367 | 131 | 92 |
| *zeb1* | Zinc finger e-box binding homeobox 1 | Repression of interleukin-2 function | GCAGAGACCAGCGGCATGTAA, ATACGAGTGCCCCAACTGTAAAA | *I. punctatus*; XM_017483097 | 156 | 89 |

### 2.7. Ethology Analysis

Ethological observations of the fish (including agonistic behavior, air breathing, escape attempts, stereotypy, and swimming activity) were conducted via video recordings in both experiments to determine possible effects of the Fe-DTPA on the fish. The observations were completed on days 8, 18, and 28 of the experiments for 20 min following a standard rotating box model to ensure a systematic recording structure over the periodic recording events. In this model, three tanks from different groups were recorded at the same time, rotating in order from the following session. At the first recording session, no fertilizer was used in order to provide a reference for each group in subsequent sessions. In this way, differences that were caused by the individual composition of the fish stocking could be better identified, if necessary. The first five minutes in each video recording were not evaluated to exclude anthropogenic influences (camera placement or movement in front of the tanks, etc.). An ethogram, according to Van de Nieuwegiessen et al. [44], was slightly modified, and used to evaluate the behavior (Table 2). The fish activity was calculated based on all visible swimming and resting fish within a five-minute interval, followed by counting all swimming and resting individuals and comparing these numbers to the total visible fish.

**Table 2.** Ethogram, direct observations and video recording [44].

| Behavioral Element | Definition |
|---|---|
| Agonistic behavior | Chasing or biting a fish or being chased by or bitten by another fish. |
| Air breathing | The fish moves to the water surface and takes a gulp of air. This was checked by escaping air from the gills of the fish when it was swimming back to the bottom of the tank. |
| Escape attempts | The fish moves to the water surface and its head emerges from the water surface past its gill cover. |
| Resting | Moving passively through the water or lying still at the bottom of the tank. |
| Stereotypic behavior | Continuous and compulsive swimming in a fixed, repetitive pattern. |
| Swimming | Displacement of the body, while browsing, moving, eating and air-breathing. |

### 2.8. Statistics

The statistical analysis was conducted using the Statistical Package for the Social Sciences (SPSS), version 27.0 (IBM Corp., 2020, Armonk, NY, USA). First, the normal distribution and homogeneity of variances of the data were tested by Shapiro and Levene tests. For normal distributed data, a one-way analysis of variance (ANOVA) was conducted. If the data were not normally distributed, a Kruskal–Wallis test was used. If the Levene test indicated inhomogeneous variances, a one-way Welch´s ANOVA was performed. For significances ($p < 0.05$), the post-hoc tests Tukey HSD (ANOVA), Dunnett-T3 (Kruskal–Wallis) or Games–Howell (Welch ANOVA) were conducted. The obtained multiplex qPCR data were analyzed using the RealTime PCR Analysis Software v. 4.5.2 (Standard BioTools) and normalized against the geometric mean of three suitable reference genes (*rna18s, actb, gapdh*). Those normalized copy numbers of the target genes were then evaluated using one-way ANOVA.

## 3. Results

### 3.1. Water Analyses

The mean physicochemical water parameters are given in Table 3. The water temperature and oxygen saturation were constant during both of the experiments ($p > 0.05$). The EC values fluctuated among the treatment groups and showed a maximum within the highest used Fe-DTPA concentrations. These high concentrations (A-FeDTPA-3; B-FeDTPA-12) were significantly different from both the lower concentrations (A-FeDTPA-0.75; B-FeDTPA-3) and the control groups (FeDTPA-0) ($p < 0.05$). In addition, the redox potential decreased mostly with increasing Fe-DTPA. The treatments A-FeDTPA-0.75, A-FeDTPA-3; B-FeDTPA-3, B-FeDTPA-12 differed significantly from respective controls (FeDTPA-0) or (in Exp. A) with lower Fe-DTPA concentration. During Exp. B, the pH values were significantly different between B-FeDTPA-0 and B-FeDTPA-3. In both the experiments,

the $NH^{4+}$ concentrations differed significantly between FeDTPA-0 and FeDTPA-3 ($p < 0.05$). Furthermore, significant differences were found in the different $Fe^{3+}$ and $Fe^{2+}$ contents, which was intended. Thus, for Fe total, there was a significant increase at each increment between treatment groups.

### 3.2. Growth and Mortality

The initial weights and lengths of the fish were similar ($p > 0.05$) across all the groups within each experiment (Table 4). The final weights, final lengths, and CF were also similar ($p > 0.05$). During Exp. A, there was no mortality. In Exp. B, on the other hand, a few losses occurred ($p > 0.05$) in the tanks, but not regularly. Instead, these fish left their tanks by jumping and died. The percentage of mortalities as well as the actual numbers of fish that died are given in Table 4.

### 3.3. Fish Behavior

There were no significant changes in the behavior of the fish from the different treatment groups after addition of the Fe-DTPA in either experiment. In Exp. A, only one significant difference was found in the agonistic behavior between the FeDTPA-0.75 and FeDTPA-3 groups, but prior to Fe-DTPA exposure (Table 5). Nevertheless, minor effects were observed in all treatment groups during both the experiments. For instance, increasing air breathing activity was observed in Exp. A, while in Exp. B, a slight increase in agonistic behavior led to elevation in skin lesions. Regarding the skin lesions, it was found that the number increased with the course of the experiment. An exception was the B-Fe-DTPA-12 group, where the number decreased throughout the experiment. Significant differences were found in the B-Fe-DTPA-3 group between day 11 and day 21 as well as day 11 and day 32. Regarding lesion area, there was a significant difference in the B-Fe-DTPA-3 group between days 11 and 32.

### 3.4. Leucogram

In Exp. B, the total amounts of leucocytes were detected. The percentage distribution of leucocyte cell types is shown in Table 6. The distribution of lymphocytes ranged from $48.92\% \pm 3.27\%$ to $73.63\% \pm 2.65\%$ during the exposition trial. Monocytes were detected between $14.18\% \pm 1.49\%$ and $27.99\% \pm 3.52\%$. A class distribution of known types of granulocytes of the African catfish were calculated as a percentage fraction of the total cell count and ranged from $0.49\% \pm 0.28\%$ to $3.22\% \pm 0.53\%$ (neutrophile meta-granulocytes), $1.12\% \pm 0.34\%$ to $2.64\% \pm 1.18\%$ (neutrophile meso-granulocytes), and $26.63\% \pm 0.64\%$ to $5.54\% \pm 1.12\%$ (mature-neutrophils). Scattered eosinophils were also counted with day 21 onwards. At day 11, major cell types of leucocytes (lymphocytes and monocytes) as well as types of upcoming neutrophilic granulocytes did not differ significantly among the treatments ($p > 0.05$). Only the cell distribution of mature neutrophils was significantly different between the B-FeDTPA-12 and the B-FeDTPA-0 groups, followed by a significant difference in lymphocytes between the B-FeDTPA-3 and the B-FeDTPA-0 groups after 21 days of exposure ($p < 0.05$). Different amounts of monocytes were found between day 11 and day 32 (ANOVA, $p > 0.1$, Dunnet-T3). Over the entire experiment (32 days), cell distribution of mature neutrophils as well as cell types of its precursor states (neutrophilic meta/meso granulocytes) decreased significantly in all three groups ($p < 0.05$).

### 3.5. Histological Assessment

In Exp. B, the histopathological examination revealed minimal (B-FeDTPA-0 and B-FeDTPA-3, both G1) to mild (B-FeDTPA-12, G2) accumulations of $Fe^{3+}$ within the liver tissue (Figure 1a–c) after 32 days. The cells of the gill filaments of each group showed no Fe-inclusion. However, hypertrophic swelling occurred in the lamellae of the gill filaments, particularly in the B-FeDTPA-3 (G2) and the B-FeDTPA-12 groups. This was not remarkable (G0) in the control group. Hyperplasia was also detected in the B-FeDTPA-12 group after 32 days, which was therefore classified as moderate (G3) (Figure 1d–f).

**Table 3.** Mean water parameters (±standard deviation (SD); n = 22) in Exp. A (fingerlings) and Exp. B (juveniles). In each experiment, means in a row followed by different superscripts are significantly different ($p < 0.05$) according to Kruskal–Wallis and Dunnett T3, respectively.

| Parameters | | Exp. A | | | | | | Exp. B | | | | | |
|---|---|---|---|---|---|---|---|---|---|---|---|---|---|
| | | FeDTPA-0 | ±SD | FeDTPA-0.75 | ±SD | FeDTPA-3 | ±SD | FeDTPA-0 | ±SD | FeDTPA-3 | ±SD | FeDTPA-12 | ±SD |
| Temp. | (°C) | 29.80 | 1.06 | 29.70 | 0.78 | 29.50 | 1.00 | 29.00 | 1.10 | 28.00 | 0.61 | 28.90 | 0.76 |
| DO | (mg/L) | 8.20 | 0.27 | 8.10 | 0.18 | 8.10 | 0.19 | 7.10 | 0.39 | 6.80 | 0.52 | 6.90 | 0.36 |
| EC | (µS/cm) | 523.80 [a] | 21.70 | 534.00 [a] | 21.86 | 545.30 [b] | 39.35 | 986.80 [a] | 86.69 | 958.70 [a] | 77.87 | 1058.20 [b] | 129.11 |
| Redox | (mV) | 190.00 [a] | 15.89 | 151.80 [b] | 32.56 | 122.90 [c] | 51.21 | 169.00 [a] | 15.12 | 126.50 [b] | 27.69 | 128.80 [b] | 28.12 |
| pH | | 8.70 | 0.07 | 8.70 | 0.10 | 8.70 | 0.08 | 6.40 [a] | 0.74 | 7.00 [b] | 0.50 | 6.70 [c] | 0.65 |
| $NO_3^-$ | (mg/L) | 12.31 | 2.63 | 13.02 | 3.72 | 13.92 | 3.95 | 201.60 | 78.74 | 174.95 | 79.09 | 185.92 | 78.13 |
| $NO_2^-$ | (mg/L) | 0.00 | 0.00 | 0.01 | 0.01 | 0.01 | 0.00 | 0.28 | 0.38 | 0.27 | 0.30 | 0.49 | 0.48 |
| $PO_4^{2-}$ | (mg/L) | 1.00 | 0.37 | 0.58 | 0.13 | 0.65 | 0.36 | 6.78 | 3.37 | 4.72 | 2.39 | 5.84 | 2.58 |
| $NH_4^+$ | (mg/L) | 0.10 | 0.09 | 0.07 | 0.07 | 0.08 | 0.10 | 2.03 [a] | 1.92 | 0.39 [b] | 0.60 | 0.84 [c] | 0.94 |
| $Fe^{3+}$ | (mg/L) | 0.01 | 0.00 | 0.00 | 0.00 | 0.01 | 0.00 | 0.02 | 0.03 | 0.02 | 0.02 | 0.03 | 0.03 |
| $Fe^{2+}$ | (mg/L) | 0.06 [a] | 0.03 | 0.58 [b] | 0.22 | 2.36 [c] | 0.98 | 0.07 [a] | 0.03 | 2.24 [a] | 1.21 | 9.00 [b] | 4.94 |
| Fe total | (mg/L) | 0.08 [a] | 0.02 | 0.59 [b] | 0.22 | 2.37 [c] | 0.99 | 0.09 [a] | 0.01 | 2.25 [b] | 1.18 | 8.99 [c] | 4.90 |

**Table 4.** Mean growth (±standard deviation (SD)) and mortality of *Clarias gariepinus* exposed to different Fe-DTPA concentrations. No significant differences were found.

| | Exp. A | | | Exp. B | | |
|---|---|---|---|---|---|---|
| | FeDTPA-0 | FeDTPA-0.75 | FeDTPA-3 | FeDTPA-0 | FeDTPA-3 | FeDTPA-12 |
| Initial weight (g) | 0.229 ± 0.04 | 0.217 ± 0.03 | 0.171 ± 0.04 | 271.43 ± 12.75 | 272.82 ± 20.21 | 256.87 ± 19.57 |
| Final weight (g) | 2.318 ± 0.98 | 2.558 ± 1.03 | 2.931 ± 0.83 | 270.40 ± 56.80 | 315.88 ± 70.93 | 286.31 ± 45.54 |
| Initial length (cm) | 3.1 ± 0.1 | 3.1 ± 0.1 | 3.0 ± 0.1 | 30.5 ± 1.8 | 32.67 ± 0.5 | 30.8 ± 2.8 |
| Final length (cm) | 6.3 ± 1.3 | 6.6 ± 1.3 | 7.0 ± 0.7 | 33.7 ± 1.8 | 33.5 ± 2.2 | 33.3 ± 2.3 |
| Total feed input (kg) | 0.03907 | 0.03987 | 0.04702 | 2.022 | 2.038 | 1.971 |
| Condition factor | 0.79 ± 0.03 | 0.84 ± 0.03 | 0.79 ± 0.01 | 0.81 ± 0.11 | 0.77 ± 0.05 | 0.8 ± 0.05 |
| Mortality in % (n) | 0 (0) | 0 (0) | 0 (0) | 11.1 (2) | 5.5 (1) | 5.5 (1) |

**Table 5.** The mean behavioral responses (±standard deviation) of *C. gariepinus* fingerlings (Exp. A) and juveniles (Exp. B) (both with n = 3) exposed to different amounts of Fe-DTPA.

| Video and Direct Observations | Exp. A | | | Exp. B | | |
|---|---|---|---|---|---|---|
| | FeDTPA-0 | FeDTPA-0.75 | FeDTPA-3 | FeDTPA-0 | FeDTPA-3 | FeDTPA-12 |
| Swimming activity (Mean %) day 8 | 70.70 ± 9.66 | 62.22 ± 24.11 | 77.38 ± 6.48 | 77.67 ± 8.52 | 74.88 ± 7.73 | 75.53 ± 5.03 |
| Swimming activity (Mean %) day 18 | 68.47 ± 16.82 | 75.34 ± 2.93 | 78.40 ± 6.00 | 82.40 ± 4.50 | 74.09 ± 14.70 | 83.42 ± 5.04 |
| Swimming activity (Mean %) day 28 | 69.99 ± 12.67 | 80.35 ± 11.28 | 74.42 ± 12.73 | 78.80 ± 12.65 | 79.69 ± 17.74 | 86.82 ± 13.94 |
| Escape attempts (Total frequency) day 8 | 1 ± 0.60 | 0 ± 0.00 | 1 ± 0.60 | 0 ± 0.00 | 0 ± 0.00 | 0 ± 0.00 |
| Escape attempts (Total frequency) day 18 | 1 ± 0.60 | 0 ± 0.00 | 0 ± 0.00 | 0 ± 0.00 | 0 ± 0.00 | 0 ± 0.00 |
| Escape attempts (Total frequency) day 28 | 0 ± 0.00 | 0 ± 0.00 | 2 ± 1.20 | 0 ± 0.00 | 0 ± 0.00 | 1 ± 0.60 |
| Air breathing Index (Total frequency) day 8 | 0.81 ± 0.43 | 0.52 ± 0.50 | 0.74 ± 0.25 | 6.31 ± 0.34 | 5.39 ± 0.42 | 6.30 ± 1.14 |
| Air breathing Index (Total frequency) day 18 | 1.36 ± 0.31 | 1.14 ± 0.49 | 1.48 ± 0.90 | 5.70 ± 1.18 | 7.33 ± 1.18 | 5.69 ± 1.28 |
| Air breathing Index (Total frequency) day 28 | 2.02 ± 0.43 | 4.52 ± 1.69 | 1.69 ± 1.49 | 5.48 ± 1.63 | 8.49 ± 1.81 | 6.48 ± 1.46 |
| Agonistic behavior Index (Total frequency) day 8 | 1.55 ± 0.18 | 1.02 ± 0.23 [a] | 1.62 ± 0.25 [b] | 0.15 ± 0.21 | 0.11 ± 0.15 | 0.11 ± 0.15 |
| Agonistic behavior Index (Total frequency) day 18 | 2.90 ± 1.03 | 2.43 ± 0.58 | 2.40 ± 0.57 | 0.29 ± 0.32 | 0.29 ± 0.31 | 0.44 ± 0.60 |
| Agonistic behavior Index (Total frequency) day 28 | 2.69 ± 0.61 | 2.50 ± 0.31 [a] | 1.74 ± 0.35 [b] | 0.78 ± 0.84 | 0.67 ± 0.70 | 0.39 ± 0.67 |
| Stereotypical behavior (Total frequency) day 8 | 0 ± 0.00 [a] | 0 ± 0.00 [a] | 0 ± 0.00 [a] | 0 ± 0.00 | 0 ± 0.00 | 0 ± 0.00 |
| Stereotypical behavior (Total frequency) day 18 | 0 ± 0.00 [a] | 0 ± 0.00 [a] | 0 ± 0.00 [a] | 0 ± 0.00 | 0 ± 0.00 | 0 ± 0.00 |

**Table 5.** *Cont.*

| Video and Direct Observations | Exp. A | | | Exp. B | | |
|---|---|---|---|---|---|---|
| | FeDTPA-0 | FeDTPA-0.75 | FeDTPA-3 | FeDTPA-0 | FeDTPA-3 | FeDTPA-12 |
| Stereotypical behavior (Total frequency) day 28 | $0 \pm 0.00$ [a] | $0 \pm 0.00$ [a] | $0 \pm 0.00$ [a] | $0 \pm 0.00$ | $0 \pm 0.00$ | $0 \pm 0.00$ |
| Skin lesions (n) day 11 | n.g. | n.g. | n.g. | 5 [A] | 0 [A] | 3 [A] |
| Skin lesions (n) day 21 | n.g. | n.g. | n.g. | 3 [A] | 13 [B] | 6 [A] |
| Skin lesions (n) day 32 | n.g. | n.g. | n.g. | 11 [A] | 29 [B] | 3 [A] |
| Skin lesion area (cm$^2$) day 11 | n.g. | n.g. | n.g. | $0.43 \pm 0.92$ [A] | $0.00 \pm 0.00$ [A] | $0.33 \pm 0.66$ [A] |
| Skin lesion area (cm$^2$) day 21 | n.g. | n.g. | n.g. | $0.30 \pm 0.75$ [A] | $0.65 \pm 0.56$ [B] | $0.60 \pm 1.14$ [A] |
| Skin lesion area (cm$^2$) day 32 | n.g. | n.g. | n.g. | $0.70 \pm 1.06$ [A] | $1.55 \pm 1.81$ [B] | $0.15 \pm 0.41$ [A] |

Note: Day 8 was prior to the first Fe-DTPA application. The respective Fe-DTPA concentrations were adjusted from day 10 on. The skin lesions were documented during the blood sampling. The skin lesion area is given as mean of nine sampled fish per day. Data with significant differences ($p < 0.05$) have been marked with different superscripts according to Kruskal–Wallis and Dunnett T3 test or ANOVA and Tukey HSD. Lower-case superscripts compare groups per day, upper-case superscripts compare the sampling days in a single group (only in skin lesions).

**Table 6.** Differential blood count in Exp. B. The distribution of leucocyte types of *C. gariepinus* juveniles during the exposure to different concentrations of Fe-DTPA (means $\pm$ standard deviation, n = 3). Means in a row followed by different superscripts are significant different ($p < 0.05$), according to Welch-ANOVA, Games-Howell. Small superscripts are significances between treatment groups within a sampling event; large superscripts are significances of particular groups over the entire study period. N.g. = not given.

| Cell Types (%) | Day 11 (after 24 h) | | | Day 21 | | | Day 32 | | |
|---|---|---|---|---|---|---|---|---|---|
| | FeDTPA-0 | FeDTPA-3 | FeDTPA-12 | FeDTPA-0 | FeDTPA-3 | FeDTPA-12 | FeDTPA-0 | FeDTPA-3 | FeDTPA-12 |
| Lymphocytes | $52.28 \pm 1.73$ [A] | $52.91 \pm 2.17$ [A] | $48.92 \pm 3.27$ [A] | $58.78 \pm 2.48$ [a] | $73.63 \pm 2.65$ [b] | $58.41 \pm 10.93$ | $65.27 \pm 7.39$ [B] | $65.09 \pm 4.50$ [B] | $57.60 \pm 8.77$ [B] |
| Monocytes | $19.59 \pm 2.84$ | $17.22 \pm 7.18$ | $19.53 \pm 2.82$ | $20.86 \pm 7.81$ | $14.35 \pm 1.49$ | $20.32 \pm 7.12$ | $26.40 \pm 6.62$ | $23.12 \pm 1.00$ | $28.11 \pm 3.52$ |
| Mature neutrophils | $22.35 \pm 1.12$ [aA] | $24.78 \pm 7.96$ [A] | $26.63 \pm 0.64$ [bA] | $16.90 \pm 4.74$ | $9.03 \pm 2.44$ | $17.85 \pm 5.72$ | $6.17 \pm 1.39$ [B] | $7.53 \pm 2.51$ [B] | $5.54 \pm 1.12$ [B] |
| Neutrophile meta-granulocyte | $3.22 \pm 0.53$ [A] | $2.56 \pm 1.93$ [A] | $2.74 \pm 0.09$ [A] | $0.86 \pm 0.55$ [B] | $1.09 \pm 0.63$ [B] | $1.22 \pm 0.20$ [B] | $0.49 \pm 0.28$ [B] | $0.64 \pm 0.44$ [B] | $0.63 \pm 0.49$ [B] |
| Neutrophile meso-granulocyte | $2.48 \pm 0.21$ [A] | $2.64 \pm 1.18$ [A] | $2.16 \pm 0.96$ [A] | $2.39 \pm 1.39$ | $1.80 \pm 1.09$ | $2.14 \pm 0.85$ | $1.12 \pm 0.34$ [B] | $1.81 \pm 0.58$ [B] | $1.58 \pm 0.71$ [B] |
| Eosinophils | n.g. | n.g. | n.g. | $0.19 \pm 0.33$ | $0.06 \pm 0.10$ | n.g. | $0.48 \pm 0.52$ | $1.79 \pm 2.03$ | $0.37 \pm 0.37$ |

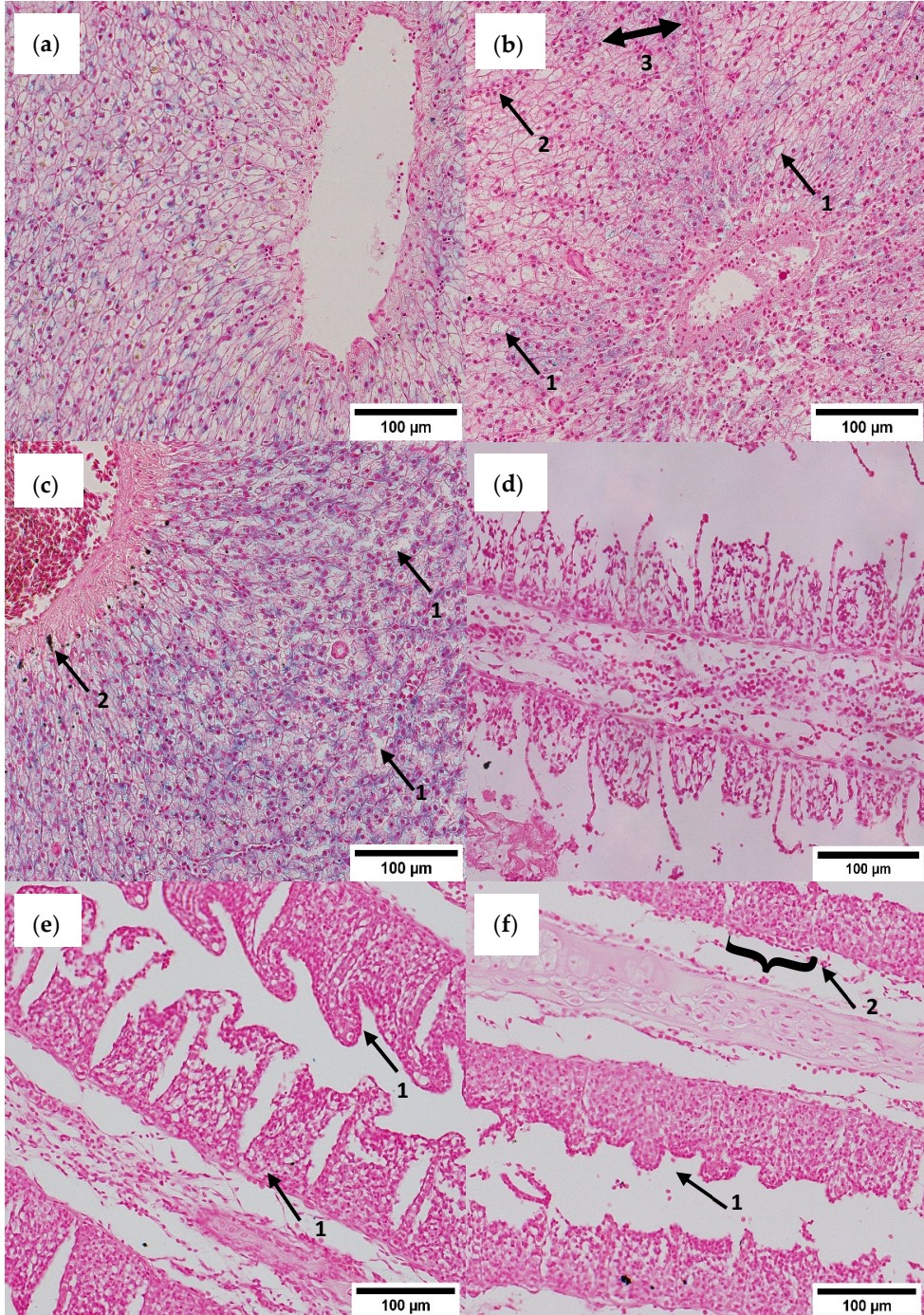

**Figure 1.** Histopathological evaluation of liver and gill tissue of *C. gariepinus* exposed to different Fe-DTPA concentrations in Exp. B after 32 days; (**a**) liver cells of the B-FeDTPA-0 group with noticeable Fe-accumulation; (**b**) liver cells of the B-FeDTPA-3 group with minimal Fe-accumulation near the vascular arteria and lipid droplets occurring next to hepatocytes (arrow 1), sinusoids congested with red blood cells (arrow 2), and hepatic plate (arrow 3); (**c**) liver cells of the B-FeDTPA-12 group with mild Fe-accumulation around vascular arteria tissue with lipid droplets (arrow 1) and gall secrets (arrow 2), central vein congested with red blood cells; (**d–f**) gill filaments of the B-FeDTPA-0, B-FeDTPA-3, and B-FeDTPA-12 groups without remarkable inclusions or necrosis. Hypertrophy, epithelial detachments, and secondary lamellae hyperemia of gill filaments occurred in the B-FeDTPA-3 and B-FeDTPA-12 groups (**e**,**f**) (arrows 1), and hyperplasia (arrow 2) was also detected in B-FeDTPA-12 (**f**). Tissues stained with Prussian blue regarding to Perls [45].

### 3.6. Gene Expression Profiling

We established a panel of 34 *C. gariepinus*-specific qPCR assays to assess the effects of different Fe-DTPA concentrations on the expression of various immune- and stress-related parameters in the head kidney of individuals compared to a control group without Fe-DTPA (Figure 2). We also designed assays for the reference genes *rna18s*, *rpl*, *actb*, and *gapdh*, and profiling revealed minor transcriptional alterations to varying Fe concentrations.

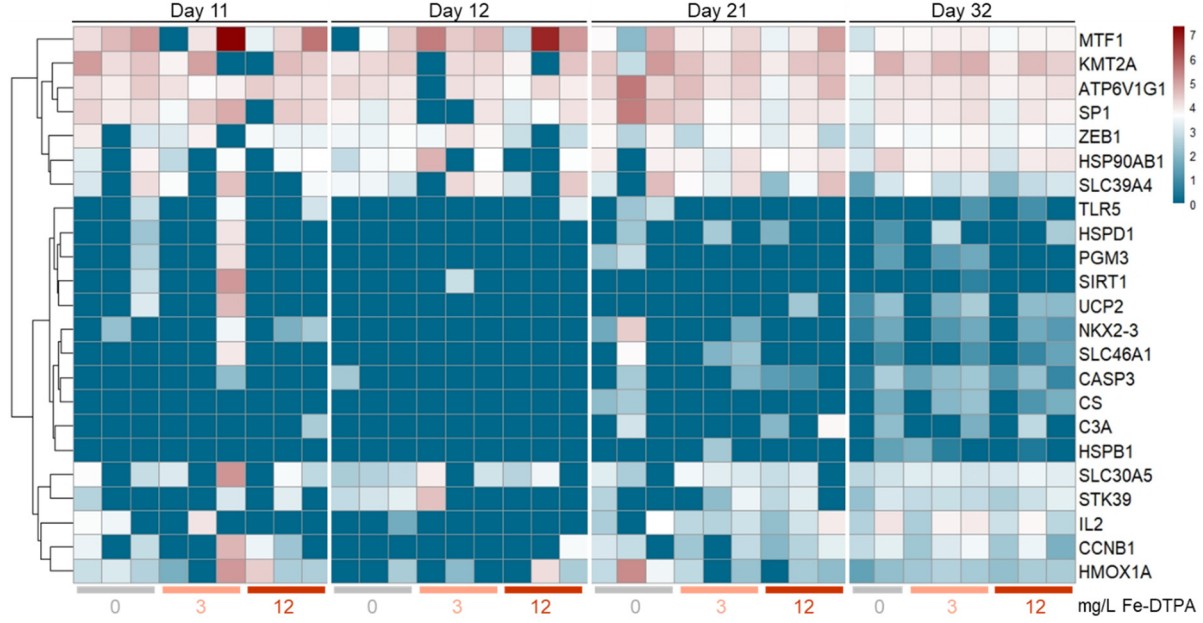

**Figure 2.** Hierarchical clustering of log10-transformed transcript numbers measured in the head kidney of individual African catfish exposed to Fe-DTPA concentrations given below the heatmap; the sampling time points (days) are indicated above the heatmap. The transcripts quantified are listed as gene symbols on the right margin; transcript levels are colored according to the scale on the right.

Twelve of the selected target genes (*casp8*, *cp*, *hsf*, *igf1*, *il4*, *il6*, *nr3c1*, *oser1*, *osgin2*, *st8sia4*, and *tnf*) were not detectable or revealed copy numbers at negligibly low levels. The transcript levels of seven genes (*c3a*, *hspb1*, *nkx2-3*, *pgm3*, *sirt1*, *slc46a1*, and *tlr5*) remained low across all groups examined, whereas the levels of four other genes (*casp3*, *cs*, *hspb1*, and *ucp2*) increased from low to partly moderate transcript numbers after one and/or two weeks of exposure to Fe-DTPA, but not significantly. For many genes with a moderate basal expression (mostly *atp6v1g1*, *hsp90ab1*, *kmt2a*, *mtf1*, *slc39a4*, *sp1*, and *zeb1*), the transcript levels varied between individuals of one cohort, revealing not significant gene expression differences across experimental groups and time points. Altogether, the qPCR data revealed a poor, not significant response in the 23 detectable genes in *C. gariepinus* exposed to Fe.

## 4. Discussion

### 4.1. Water Quality

The African catfish tolerates a wide range of water quality parameters, i.e., a temperature between 18–32 °C, albeit the optimal temperature for development and growth ranges between 27–29 °C [46]. A pH between 5.0 and 9.0 meets the requirements for this species [46]; in aquaponics, however, the pH is usually lower (approx. 6.5–7.0) because of more efficient bacterial activity and nutrient uptake of the plants [47]. DO levels were recommended between 3–6 mg/L [46], but higher levels, up to approx. 9.0 mg/L, are also considered to be adequate [48]. EC values between 100 and approx. 2000 μS/cm has been reported to be well tolerated by the fish [48,49]. Concentrations of $NO_3$, $NO_2$, and $NH_4(NH_3)$ had been shown to be still tolerated in ranges up to 697.5 mg/L, 0.6 mg/L, and 20.46 mg/L (>0.34 mg/L), respectively [50–53]. Oxidative regimes with a positive redox

potential between, for instance, +83.4 and +151.7 mV were recommended for bacterial activity in aquaponic systems and can be tolerated by *C. gariepinus* [54,55].

In the present study, the temperatures were between 28–29 °C, the pH was between 6.0–8.0, the DO levels ranged from 7 to 8 mg/L, EC were between approx. 500–1060 μS/cm, and redox potentials were between 122 and 190 mV; all parameters therefore fit to the optimal or an adequate range for the African catfish. In both the experiments, the redox potential decreased along with higher fertilization but remained in a positive (oxidative) condition. As the decreasing redox potential was seen to be non-lineal in Exp. B, the deactivation of redox active metal ions ($Fe^{2+}$, $Fe^{3+}$) by used DTPA-chelator can be assumed to have created a buffering effect to the redox potential [56,57].

The Fe concentrations were taken constantly through a short time monitoring system on a defined level between 0.75 mg/L and 3.0 mg/L in Exp. A and 3.0 mg/L and 12.0 mg/L in Exp. B, respectively. In some cases, the treatment groups exceeded the general needs of plants in an aquaponic system (2–5 mg/L) but did not indicate a strong impact on the fish. Other monitored chemical water parameters (pH, $O_2$, $NH_4$, $NO_2$, $NO_3$, and $PO_4$) were not affected by Fe-DTPA concentrations or seemed to be influenced by the treatment.

### 4.2. Growth Performance and Mortality

First, it should be mentioned that neither of the experiments were designed for the most efficient growth performance. Both trials were conducted in aquarium systems, which only allowed a limited supply of nutrients, ensuring the system stability. As a result, the fish in all treatment groups grew by only a few grams. The partially large differences between the mean values are due to the fact that sometimes smaller, and sometimes larger fish were taken for sampling, and at the end of the experiment, only four or five fish remained for the mean value calculation. This indicates a strong heterogeneity of growth, which can occur particularly in the case of low feed distribution [58]. The mortality in African catfish fingerlings (1.15–1.63 g) have been described in a range between 4.3–20.0% [59,60], whereas for juveniles (102–288 g), the range is between 2.0–4.8% [53,61]. In the present study, it was in a similar range (between approx. 5.5% and 11.1%). No influence of the different Fe concentrations was found, since on the one hand, only jumpers died, but on the other hand, the control was in a similar range.

### 4.3. Physiological and Behavioral Responses

Unaffected leukocyte distributions of African catfish (with weights of approx. 320 g) were described with lymphocytes at 55.0%, monocytes at 3.2%, and neutrophils at 43.4% [62], and with lymphocytes at 69.0%, monocytes at 3.0%, neutrophils/heterophils at 25.0%, and eosinophils at 3.0% [63].

In the present study, we found a significantly lower number for all three types of neutrophilic granulocytes along with a significantly increased number of lymphocytes at the end of the experiment (after 32 days) in relation to the initial sampling (after 11 days) in all treatments. The number of monocytes also tended to increase ($p < 0.1$) over time. According to Clauss et al. [39], a lower presence of mature neutrophils in the peripheral blood, along with a slightly trend of a monocytosis, which was seen in the present study, is a hint for a starting inflammatory reaction. As the higher phagocytic activity reduces number of mature neutrophils, proliferation and formation of large numbers of new neutrophilic meso- and meta-granulocytes indicate an ongoing immune response. Higher rates of matured neutrophils indicate the remission of a previous immune response as all pathogen matter has eliminated. A potential inflammatory response in the present study was presumably a result of increased numbers of skin lesions following elevated agonistic interactions among the fish rather than a response to Fe concentrations in the water. In the present study, the cell counts of monocytes were generally higher, whereas neutrophils were lower (also without Fe-DTPA, respectively) in comparison to the reference values above. A distinct effect of the added Fe-DTPA is therefore questionable.

Van de Nieuwegiessen et al. [44] described agonistic behavior of juvenile African catfish (weighing between 10 to about 100 g) at 0–12 (lesion number/fish/h); furthermore, Van de Nieuwegiessen et al. [61] described agonistic behavior in African catfish (about 102–288 g) at 0–8. Both of these studies were under different stocking densities. The significant differences regarding the agonistic behavior in the present experiments should be interpreted with caution. All values are within the normal range and may easily have been influenced by, for example, group composition [64]. Comparing the significant data from day 8 (without Fe-DTPA) of Exp. A with the following days, this becomes even more evident.

The average number of skin lesions in African catfish was reported between 1.4–2.0 [64] or 0.8–2.4 [50]. In the present study, the number of skin lesions ranged between approx. 0.0–1.6, representing the normal range. More interesting is that in the B-FeDTPA-12 group, contrary to the development of the other two groups, the number of lesions slightly decreased. This could have resulted from a change in behavior upon the increased iron concentration but requires further investigation. Our analysis of agonistic behavior (frequency) cannot confirm this, but it also does not relate to the intensity of agonistic interactions. In any case, no negative influence on fish welfare or health was exerted by this.

*4.4. Histopathology and Gene Expression Profiling*

Stathopoulou et al. [1] tested Fe-DTPA (3.61 mg/L) on rocket (*Eruca vesicaria*) and Nile tilapia (*Oreochromis* spp.). Several organs showed minimal changes such as granulomas, lipid accumulation in the liver cells, and macrosteatosis of the pancreatic islets that are caused by the Fe-DTPA addition. Mild histological alterations, such as hyperplasia and telangiectasia, were found in the gills.

The histopathological results of the present study revealed changes in the liver tissue and the gills, especially under concentrations of 12.0 mg/L Fe-DTPA. The most important changes were $Fe^{3+}$ accumulations, an incipient inclusion of lipid vacuoles in the liver, as well as hypertropia/hyperplasia of the primary lamellae, epithelial detachments, and secondary lamellae hyperemia of the gills. The recent score values of the assessed liver and gill tissue ranged from 0 (not remarkable) to 3 (moderate), slightly higher than Stathopoulou et al. [1], who used the same rating system with grades 1 (minimal) to 2 (mild) for liver and gills. According to Stathopoulou et al. [1], fat deposition in the liver is affected by the dietary lipid content but can also be a reaction to intoxication. Whether Fe-DTPA at the tested concentrations can lead to severe lipid accumulation or fatty liver need to be tested in longer-term studies. However, over a period of up to three weeks, as demonstrated in the present study, the documented alterations were not detrimental to fish health to a greater extent.

The gene expression profiling revealed no significant differences between the head kidney samples from the different treatment groups. A number of gene transcripts were undetectable with our qPCR-assay panel, but this was not an unexpected finding. Some of the selected genes are only expressed in a tissue-specific manner (*st8sia4*, for instance, is mainly present in cells of the gonads and nervous system of African catfish) and were weakly detectable in the head kidney samples in our pretests. Other genes (including *il6* and *tnf*) are known to be induced in response to acute stimuli of a given intensity [65]. Obviously, the experimental stimulus (namely the tested concentrations of Fe-DTPA) was not intense enough to provoke the induction of acute-phase and other genes measured in the present study. However, we cannot exclude that distinct genes, which were not integrated as targets in our analysis, had been modulated in their expression by Fe-DTPA exposition. Future investigations might consider holistic approaches [66,67] to identify previously unknown indicators.

**5. Conclusions**

Growth and mortality of African catfish were unaffected by Fe-DTPA in the tested concentrations up to 12.0 mg/L. The ethological analysis also indicated no dependency

of the behavior on Fe-DTPA treatments. Nevertheless, since minor accumulation of $Fe^{3+}$ in liver tissue as well as histomorphological changes in gill tissue occurred in particular at 12.0 mg/L but were negligible at 3.0 mg/L, and since the fertilization recommendation of Fe in hydroponic cultures is 4.0 mg/L, we recommend a lower addition to aquaponics with African catfish. Long-term studies should continue to clarify whether there can also be a stronger influence on or damage to the animals over longer exposure.

**Author Contributions:** The authors of this article made the following contributions: conceptualization, M.-C.H. and B.B.; methodology, M.-C.H., B.B. and A.R.; validation, M.-C.H., B.B. and A.R.; formal analysis, M.-C.H.; investigation, M.-C.H., B.B., A.R. and J.A.N.; resources, M.-C.H. and A.R.; data curation, M.-C.H., A.R.; writing—original draft preparation, M.-C.H. and B.B.; writing—review and editing, M.-C.H., B.B., H.W.P. and A.R.; visualization, M.-C.H. and A.R.; supervision, B.B. and H.W.P.; project administration, B.B.; and funding acquisition, H.W.P. All authors have read and agreed to the published version of the manuscript.

**Funding:** This study was funded by the project "Performance and process water management in commercial (integrated) aquaculture systems with African catfish (*Clarias gariepinus*) in Mecklenburg-Western Pomerania" (MV-II.1-LM-007: 139030000103, EMFF: 7302).

**Data Availability Statement:** The datasets generated and/or analyzed during the current study are not publicly available, although they are available from the corresponding author on reasonable request.

**Acknowledgments:** We thank Markus Kipp and Frauke Winzer (Institute for Anatomy, University of Rostock) for histological sample preparation and assistance and Julian Krinitskij (FBN) for his assistance with the gene-expression profiling. We thank Erwin Berchtold, Leopold Hummel, Lisa Carolina Wenzel, Lu Xu, and Laura Ballesteros Redondo (Department of Aquaculture and Sea-Ranching, University of Rostock) for their assistance during samplings.

**Conflicts of Interest:** The authors declare no conflict of interest. The funders had no role in the design of the study; in the collection, analysis, or interpretation of the data; in the writing of the manuscript; or in the decision to publish the results.

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
