# Peer review of "Effects of Fe-DTPA on Health and Welfare of the African Catfish Clarias gariepinus (Burchell, 1822)"

_water, doi:10.3390/w15020299_

Round 1

Reviewer 1 Report

Positive comments:

1. The topic of the scientific research is current, as it concerns Effects of Fe-DTPA on Health and Welfare of the African Catfish;

2. The object of the study is African catfish, which is an economically valuable fish and is cultivated in an aquap system, combined with plant species;

3. The working hypothesis is correctly formulated, since indeed ferrications are less diffusible than ferrocations;

4. The source of iron to overcome iron deficiency in African catfish is properly selected, as iron from plant cultivation alone is not sufficient. This is because for plants iron is a micronutrient and for animals it is a macronutrient;

5. Detailed studies have been made regarding the physicochemical parameters of the water and the biochemical-genetic and histological parameters of the African catfish;

6. An adequate discussion has been made, including own and world authors' research, which is useful for the scientific branch;

7. At the end of the article, a conclusion is drawn, which, however, is integrated with recommendations for practice.

Negative comments and recommendations:

  1. For better visibility, I recommend that the photos to figure 1 be enlarged by 10-15%;

2. In the Discussion section, a more in-depth discussion could be made regarding the biochemical effects of Fe-DTPA on fish metabolism, and in particular on iron-containing metalloprotein/ferritin exchange, iron-containing superoxide dismutase, the cytochrome system, hemoglobin and myoglobin, etc./;

3. Conclusions and recommendations for practice are integrated in the Conclusions section. More specific recommendations could be formulated that would be valuable to fish farmers cultivating African catfish in hydroponics integrated with plant species cultivation;

4. I recommend that the authors conduct further biochemical studies with Fe-DTPA supplementation on the effect on iron-containing metalloproteins, particularly iron-containing superoxide dismutase, which is relevant to oxidative stress in fish. Similar studies would also be interesting on iron cytochrome oxidoreductases, since iron there changes reversibly from ferri- to ferroform.

Author Response

Author Comments to the reviewer report:

  1. Are the results clearly presented? Can be improved

#All results were presented in the tables and figures of our study. Figure 1 is shown at the same magnification and wide enough to see the details described.

2. Are the conclusions supported by the results? Can be improved

# Our conclusion addresses the results of our study, focusing on key findings that are relevant to the application of the Fe-DTPA. 

Reviewer negative Comments

 1. For better visibility, I recommend that the photos to figure 1 be enlarged by 10-15%;

#1 As mentioned earlier, the magnification of Figure 1 is large enough to see the different coloration of the tissues by different Fe-DTPA and different alterations described in our text. We do not recommend a higher magnification of this figure.

2. In the Discussion section, a more in-depth discussion could be made regarding the biochemical effects of Fe-DTPA on fish metabolism, and in particular on iron-containing metalloprotein/ferritin exchange, iron-containing superoxide dismutase, the cytochrome system, hemoglobin and myoglobin, etc./;

2# We were unable to determine other biochemical effects (enzyme activity, ferritin, transferrin) with sufficient accuracy or at all within the scope of our study. Unfortunately, our primers for the determination of transmutase, transferrin, and ferritin activity failed. We therefore do not wish to comment on a presumed effect of Fe-DTPA on these mechanisms, as these were not investigated. We have therefore dispensed with this point of discussion.

3. Conclusions and recommendations for practice are integrated in the Conclusions section. More specific recommendations could be formulated that would be valuable to fish farmers cultivating African catfish in hydroponics integrated with plant species cultivation;

3# We have tested the compatibility of Fe-DTPA on African catfish in RAS systems and concluded that a lower dosage below 4.0 mg/L in aquaponic systems is safe for African catfish. However, we do not want to commit ourselves to a specific value, as this has not been investigated (e.g. in a series of measurements).

4. I recommend that the authors conduct further biochemical studies with Fe-DTPA supplementation on the effect on iron-containing metalloproteins, particularly iron-containing superoxide dismutase, which is relevant to oxidative stress in fish. Similar studies would also be interesting on iron cytochrome oxidoreductases, since iron there changes reversibly from ferri- to ferroform.

4# We are eager to capture and investigate further modes of action of Fe-DTPA and metabolic processes in future studies. However, this cannot be included in the current study at this time. However, we welcome the information and will include it in our research as far as possible at the next opportunity.

Reviewer 2 Report

Review Report

This is a well done study with high novelty. In this manuscript, the author evaluate the effects on health and welfare of the African catfish, to non-lethal but aquaponic-related concentrations of the chemical chelator agent Fe-DTPA, by recording the growth, mortality, ethological indicators, leukocyte distribution, histopathological changes in liver and gills, and genetic biomarkers for each group. The results indicate that Fe-DTPA supplementation in the tested concentrations can be considered relatively harm-26 less for health and welfare of African catfish. The authors provide sufficient data and figures. While the area of research is important, some rationale sharing and experimental set up/interpretation improvements are recommended.

Comments:

1. Line 176-177: The reasons for the selection of the time points for sampling should be explained. Why did group A choose these 4 time points and why did group B choose these 5 time points? Is there a reason for sampling three consecutive days on Day 11, 12 and 13? Are they all sampled the same? Also are the different sampling times for Group A and Group B comparable?

2. Line 188: Why only sample at Day 11, 21 and 32. What about the Day 12 and 13 in the method section?

3. Line 217: A similar question. Why only sample on Day 11, 12, 21 and 32. What about day 13 in the method? And for histological analysis, what is the obvious difference between the tissues sampled on day 12 and day 11?

4. Figure 1:The graphs in Table 1 only compare the differences between the treatment groups, but not the sampling times. According to the method section, the authors sampled four time points in histological analysis--Day 11, 12, 21 and 32, so which time point was sampled in Figure 1?

5. Line 53:in your economic efficiency The word your is probably a typo.

6. Line 154:2.2 Formatting errors in heading.

7. Line 235: the gene names in Line235 and elsewhere in the text should be italicized.

8. The lowercase letter abc and the uppercase letter AB appear in a large number of tables in this manuscript; please indicate the meaning in the table notes.

Author Response

Author comments:

  1. Line 176-177: The reasons for the selection of the time points for sampling should be explained. Why did group A choose these 4 time points and why did group B choose these 5 time points? Is there a reason for sampling three consecutive days on Day 11, 12 and 13? Are they all sampled the same? Also are the different sampling times for Group A and Group B comparable?

1# The fish from experiment A were examined once a week for their growth performance. This resulted in the days 1,12, 21, and 32. lengths and weights were determined and the growth of the fish was calculated.

For the larger fish from experiment B, additional methodologies were used to track further influences of the fertilizer. For example, days 11-13 were used to collect data for an expected acute phase response in addition to the standard length and weight measurements. The methodology requires more material which the fish from experiment A did not provide. The growth data from experiment A and B are comparable to each other as they were conducted at the same time points (12th, 21st, 32nd). This was added too the script. Because we did not find evidence of an acute-phase response to the selected genes, this issue was largely excluded.

2. Line 188: Why only sample at Day 11, 21 and 32. What about the Day 12 and 13 in the method section?

2# Blood testing was limited to three days because the sample size was otherwise too large to complete. It was thus focused on the beginning, middle and end of the experiment. This was added too the script.

3. Line 217: A similar question. Why only sample on Day 11, 12, 21 and 32. What about day 13 in the method? And for histological analysis, what is the obvious difference between the tissues sampled on day 12 and day 11?

3# We collected liver and gill tissues at all time points 11th,12th,13th,21st and 32nd in experiement B and compared them. Tissue samples from 4 out of 5 days was used for genetic study as this is sufficient for the expected response of the method. The difference in the samples from day 11 and 12 is the presumption of an acute phase response to the Fe-DTPA in the first 24-96 hours after fertilization. However, this was not observed.

4. Figure 1:The graphs in Table 1 only compare the differences between the treatment groups, but not the sampling times. According to the method section, the authors sampled four time points in histological analysis--Day 11, 12, 21 and 32, so which time point was sampled in Figure 1?

4# Thank you for the kind hint. All slides shown are from day 32 and therefore comparable. This was supplemented in the required places.

5. Line 53:“in your economic efficiency ” The word “your” is probably a typo

5# was corrected

6. Line 154:2.2 Formatting errors in heading.

6# was corrected

7. Line 235: the gene names in Line235 and elsewhere in the text should be italicized.

7# the gene names have been italicized in the text

8. The lowercase letter abc and the uppercase letter AB appear in a large number of tables in this manuscript; please indicate the meaning in the table notes.

8# The letters abc and ABC were used in the tables to indicate significant differences between groups of individual days (abc) and between days of individual groups (ABC). This has been indicated at the appropriate places in the tables concerned.

#We have also optimized the script in english language.